# A trans-eQTL network regulates osteoclast multinucleation and bone mass

Marie Pereira[1,2], Jeong-Hun Ko[1,2], John Logan[2], Hayley Protheroe[2], Kee-Beom Kim[3], Amelia Li Min Tan[4], Peter I Croucher[5], Kwon-Sik Park[3], Maxime Rotival[6], Enrico Petretto[4], JH Duncan Bassett[2]*, Graham R Williams[2]*, Jacques Behmoaras[1]*

[1]Centre for Inflammatory Disease, Department of Immunology and Inflammation, Hammersmith Hospital, Imperial College London, London, United Kingdom; [2]Molecular Endocrinology Laboratory, Department of Metabolism, Digestion and Reproduction, Hammersmith Hospital, Imperial College London, London, United Kingdom; [3]Department of Microbiology, Immunology, and Cancer Biology, University of Virginia School of Medicine, Charlottesville, United States; [4]Duke-NUS Medical School, Singapore, Singapore; [5]The Garvan Institute of Medical Research and St. Vincent's Clinical School, University of NewSouth Wales Medicine, Sydney, Australia; [6]Human Evolutionary Genetics Unit, Institut Pasteur, Centre National de la Recherche Scientifique, UMR 2000, Paris, France

*For correspondence:
d.bassett@imperial.ac.uk (JHDB);
graham.williams@imperial.ac.uk (GRW);
jacques.behmoaras@imperial.ac.uk (JB)

**Competing interests:** The authors declare that no competing interests exist.

**Abstract** Functional characterisation of cell-type-specific regulatory networks is key to establish a causal link between genetic variation and phenotype. The osteoclast offers a unique model for interrogating the contribution of co-regulated genes to in vivo phenotype as its multinucleation and resorption activities determine quantifiable skeletal traits. Here we took advantage of a *trans*-regulated gene network (MMnet, macrophage multinucleation network) which we found to be significantly enriched for GWAS variants associated with bone-related phenotypes. We found that the network hub gene *Bcat1* and seven other co-regulated MMnet genes out of 13, regulate bone function. Specifically, global (*Pik3cb*$^{-/-}$, *Atp8b2*$^{+/-}$, *Igsf8*$^{-/-}$, *Eml1*$^{-/-}$, *Appl2*$^{-/-}$, *Deptor*$^{-/-}$) and myeloid-specific *Slc40a1* knockout mice displayed abnormal bone phenotypes. We report opposing effects of MMnet genes on bone mass in mice and osteoclast multinucleation/resorption in humans with strong correlation between the two. These results identify MMnet as a functionally conserved network that regulates osteoclast multinucleation and bone mass.

## Introduction

The large number of common genetic variants associated with prevalent complex diseases suggests that disease pathogenesis involves perturbations to intricate gene networks (*Furlong, 2013*). The recently proposed omnigenic model of complex traits highlights the importance of *trans*-regulated networks in understanding causative disease pathways (*Boyle et al., 2017*; *Liu et al., 2019*). One way of mapping these regulatory networks is to identify the genetic control points of gene co-expression networks in a cell type relevant to the disease of interest. In this way, expression quantitative trait loci (eQTL) studies have revealed regions of the genome that harbour sequence variants affecting the mRNA expression levels of one or more genes (*Albert and Kruglyak, 2015*). These approaches identified *trans*-eQTLs, which regulate gene expression that are often observed in clusters, also known as *trans*-eQTL hotspots (*Albert et al., 2018*; *Brem et al., 2002*; *Morley et al., 2004*; *Schadt et al., 2003*). Many of the *trans*-eQTLs discovered to date in different model organisms tend to influence the expression of multiple genes (*Albert et al., 2018*; *Bagnati et al., 2019*;

*Grundberg et al., 2012*; *Johnson et al., 2015*; *Kang et al., 2014*; *Small et al., 2011*; *Yao et al., 2017*), which suggests coordinated (network-based) regulatory mechanisms might be perturbed during disease pathogenesis. Although identification of such networks has informed our understanding of gene-gene interactions, their functional validation remains challenging.

Bone-resorbing multinucleated osteoclasts are derived from the monocyte-macrophage lineage (*Udagawa et al., 1990*). They display high metabolic activity (*Indo et al., 2013*) and regulate bone mass, structure and strength. Dysregulation of osteoclastic bone resorption is a major cause of both low and high bone mass disorders including osteoporosis (*Manolagas, 2010*), and osteopetrosis (*Sobacchi et al., 2013*), and an important contributor to the pathogenesis of Paget's disease (*Galson and Roodman, 2014*), adolescent idiopathic scoliosis (AIS) (*Liu et al., 2018*), and inflammatory diseases that affect the skeleton such as rheumatoid arthritis, ankylosing spondylitis and periodontitis (*DiCarlo and Kahn, 2011*; *McInnes and Schett, 2011*). Recent genome-wide association studies (GWAS) have identified hundreds of independent polymorphic loci associated with bone diseases in humans, including many that contain osteoclast-related genes (*Albagha et al., 2010*; *Estrada et al., 2012*; *Kemp et al., 2017*; *Kim, 2018*; *Medina-Gomez et al., 2018*; *Morris et al., 2019*; *Stahl et al., 2010*). However, only a few causal genes have been implicated in disease onset and progression, suggesting that activities of complex interacting gene networks play a crucial role in establishing and optimising bone mass and strength, and in the pathogenesis of skeletal disease (*Al-Barghouthi and Farber, 2019*).

We previously developed a rapid-throughput skeletal phenotyping pipeline that combines both structural and functional parameters (*Bassett et al., 2012a*; *Freudenthal et al., 2016*). We first applied this multi-parameter phenotyping pipeline to analyse 100 knockout (KO) mice and reported nine new genetic determinants of bone mineralisation, structure and strength (*Bassett et al., 2012a*). We then extended our studies to over 500 KO mouse lines and integrated this large-scale phenotype resource with over 1000 conditionally independent SNPs at over 500 loci that significantly associate with bone mineral density (BMD) and fracture in GWAS to provide functional evidence of causation for candidate genes (*Kemp et al., 2017*; *Medina-Gomez et al., 2018*; *Morris et al., 2019*; *Trajanoska et al., 2018*). These integrated studies demonstrated how large-scale phenotyping and genetic association projects provide systems-level platforms for gene identification in skeletal disorders. Robust systems genetics approaches also incorporate functional assays in a context-dependent cell type such that the effect of genome variation can be investigated to identify cell-specific mechanisms of disease. The osteoclast has an important advantage that, physiologically, its multinucleation capability correlates with its resorptive activity as well as with quantitative in vivo traits such as bone mass, structure and strength (*Pereira et al., 2018*). By contrast, osteoclast-rich osteopetrosis occurs due to impaired osteoclast function despite normal multinucleation and may result from mutations in (i) *TCIRG1*, *CLCN7*, *OSTM1* or *CA2* deficiency of which impair acidification, or (ii) *SNX10* and *PLEKHM1* deficiency of which impair endosome trafficking (*Sobacchi et al., 2013*). Thus, we hypothesise that osteoclast gene regulatory networks play a key role to establish and maintain optimal bone structure and strength, and are perturbed in skeletal disease.

We previously investigated the genetic determinants of macrophage multinucleation, using the inbred Wistar Kyoto (WKY) rat strain that displays spontaneous macrophage fusion (*Kang et al., 2014*; *Rotival et al., 2015*). By mapping eQTLs in fusion-competent primary macrophages, we found a *trans*-regulated gene co-expression network (190 genes) enriched for osteoclast genes that we defined as a 'macrophage multinucleation network' (MMnet) (*Kang et al., 2014*). MMnet is under the genetic control of the *Trem2* (Triggering Receptor Expressed On Myeloid Cells 2) locus and includes a hub gene, *Bcat1* (Branched chain amino acid transferase 1) that has the greatest number of connections (i.e. co-expression) with other MMnet transcripts. This network contained *trans*-regulated genes previously involved in osteoclastogenesis and we defined a new role for the most significant *trans*-eQTL (*Kcnn4*) in osteoclast multinucleation, bone homeostasis and inflammatory arthritis (*Kang et al., 2014*).

Here we validate the functional role of MMnet in the regulation of osteoclast multinucleation, resorption and bone mass and strength. We report a correlation between the effects of MMnet gene knockdown on multinucleation and resorption in vitro and the effect on bone mass in vivo. Our study illustrates how the cell- specific function of osteoclast multinucleation and resorption can be used to link genetic variation within a complex network of co-regulated genes to the clinically critical phenotypes of bone mass and strength.

## Results

### MMnet regulates adult bone homeostasis via control of osteoclast multinucleation

MMnet is a co-expression network comprising 190 osteoclast-enriched eQTLs that are regulated in trans by *Trem2* (*Kang et al., 2014*). It was generated using primary macrophages from a heterogenous rat population derived from experimental crossing of inbred Wistar Kyoto (WKY) and LEW rats. Macrophages from WKY rats display spontaneous fusion and multinucleation, whereas spontaneous macrophage fusion does not occur in Lewis (LEW) rats (*Kang et al., 2014*; *Rotival et al., 2015*). Here we show that WKY rats display increased osteoclastogenesis in vitro compared to LEW rats (*Figure 1A–B*). We therefore reasoned that WKY and LEW rats should exhibit divergent skeletal phenotypes. Skeletal analysis demonstrated that, compared to LEW rats, WKY rats have decreased bone length, bone mineral content (BMC), trabecular bone volume (BV/TV) and thickness (Tb.Th), together with reduced cortical bone thickness (Ct.Th) and mineral density (BMD) (*Figure 1C–E*). These abnormalities resulted in markedly decreased bone strength (*Figure 1D–E*). Overall, these data indicate that spontaneous macrophage and osteoclast multinucleation in WKY rats is associated with low bone mass, mineralisation and strength. Thus, these findings suggest that MMnet regulates adult bone homeostasis via its action in osteoclasts to control multinucleation.

### MMnet genes are enriched in human skeletal GWAS variants

We first determined whether MMnet eQTL genes are significantly enriched for GWAS loci associated with bone disorders in humans. In comparison with the background set of eQTLs (1448 eQTLs that were previously mapped in multinucleating macrophages [*Kang et al., 2014*], MMnet was generally enriched for bone-related GWAS loci (p=0.0085, hypergeometric test, *Table 1*) and more specifically for heel bone mineral density associated GWAS variants (*Figure 1—source data 1*), which had the most significant enrichment (20 loci, p=$4.49 \times 10^{-3}$, hypergeometric test, *Table 1*). Consistent with the decreased bone length observed in WKY rats, MMnet was also significantly enriched for body height variants (24 loci, p=$1.86 \times 10^{-2}$, hypergeometric test, *Table 1*). In addition to enrichment for GWAS loci associated with skeletal traits, 9 MMnet genes have also been implicated in the pathogenesis of rare monogenic skeletal disorders (*CTSK, FAM20C, FLNA, FN1, IDUA, IL1RN, MANBA, MET, MMP9*) (*Mortier et al., 2019*). These results demonstrate that MMnet is conserved in humans and implicated in the regulation of skeletal development and bone homeostasis (*Figure 1F*).

### BCAT1 maintains normal bone mass and strength by regulating osteoclast multinucleation and function

To validate the role of MMnet, we investigated the effect of the MMnet hub gene *Bcat1* on osteoclast multinucleation and resorption, and bone mass (*Figure 2A*). 12 week-old male *Bcat1*$^{-/-}$ mice had high bone mass characterised by increased cortical thickness (Ct.Th), trabecular bone volume (BV/TV), number (Tb.N) and thickness (Tb.Th), and reduced trabecular separation (Tb.Sp) and structural model index (SMI) (n = 8 per genotype) (*Figure 2B–C*; *Figure 2—figure supplement 1*). Biomechanical 3-point bend testing demonstrated that *Bcat1*$^{-/-}$ mice had enhanced bone strength characterised by increased yield, maximum and fracture loads and stiffness (*Figure 2C*; *Figure 2—figure supplement 1*). Furthermore, static histomorphometry showed that *Bcat1*$^{-/-}$ mice had decreased osteoclast surface (Oc.S/BS) and number (Oc.N/BS), suggesting their high bone mass phenotype results from impaired osteoclast activity (*Figure 2D*). Moreover, *Bcat1*$^{-/-}$ mice had increased amounts of retained cartilage within both trabecular and cortical bone, suggesting impaired osteoclastic resorption during bone modelling and remodelling (*Figure 2—figure supplement 1D–E*). Consistent with this, siRNA-mediated knockdown of *BCAT1* expression in human osteoclasts in vitro inhibited multinucleation and resorption by 60% (*Figure 2E–F*). By comparison, knockdown of *DCSTAMP*, a master regulator of osteoclast and macrophage membrane fusion (*Yagi et al., 2005*; *Yagi et al., 2006*), resulted in 90% inhibition of osteoclast multinucleation and resorption (*Figure 2E–F*). Taken together, these data demonstrate an important new role for BCAT1 in osteoclast formation and function, and the physiological maintenance of adult bone mass and strength.

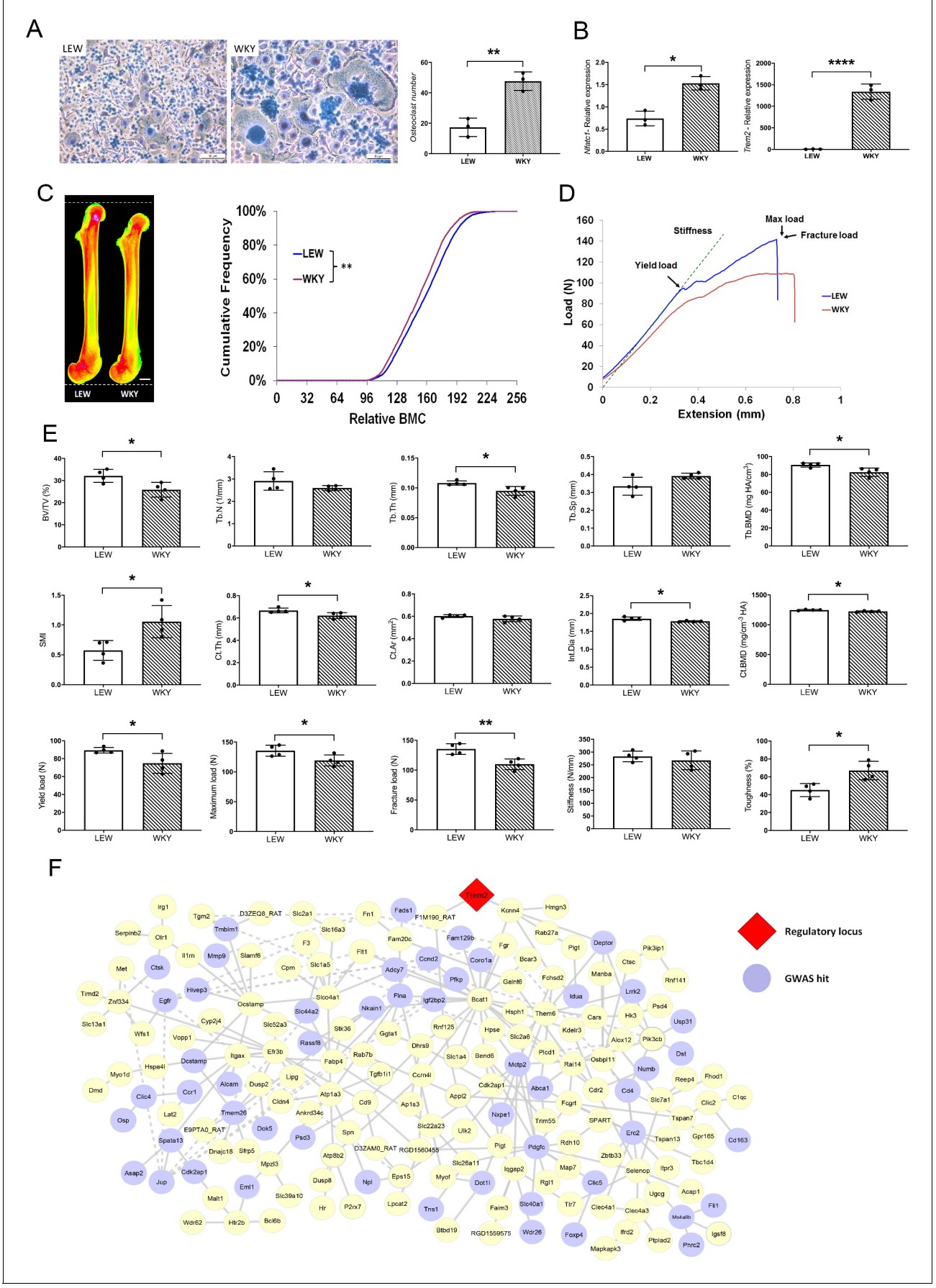

**Figure 1.** MMnet regulates adult bone homeostasis through osteoclast multinucleation. (**A**) Left: Representative images of osteoclasts from Lewis (LEW) and Wistar Kyoto (WKY) rats stained with Giemsa stain. Right: number of osteoclasts with >10 nuclei from LEW and WKY rats. (**B**) Relative mRNA expression of *Nfatc1* and *Trem2* determined by qRT-PCR following osteoclast culture (n = 3/group). (**C**) (Left) Pseudocoloured X-ray microradiography images of femurs from male Lewis (LEW) and Wistar-Kyoto (WKY) rats (low BMC blue and green and high BMC red and pink) showing decreased femur length in WKY animals (scale bar = 2 mm). (Right) Cumulative frequency histogram showing reduced BMC in WKY rats; (n = 4 \*\*p<0.01 vs. WT; Kolmogorov–Smirnov test). (**D**) Representative load displacement curves from femur three-point bend testing. (**E**) Micro-CT analysis of distal femur trabecular bone (trabecular bone volume/tissue volume (BV/TV), trabecular number (Tb.N), trabecular thickness (Tb.Th), trabecular separation (Tb.Sp), trabecular bone mineral density (Tb.BMD) and structure model index (SMI)) (Upper panel), micro-CT analysis from mid-diaphysis cortical bone (cortical thickness (Ct.Th), bone area (Ct.Ar), internal diameter (Int.Dia) and cortical bone mineral density (Ct.BMD)) (middle panel) and femur three-point-bend testing (lower panel) (Mean ± SD; n = 4 \*p<0.05, \*\*p<0.01 LEW vs WKY Student's t-test). (**F**) MMnet comprising 190 *trans*-eQTLs under the master regulatory locus (Trem2, highlighted in red). Each gene in the network is represented as a circle (node), and the genes that are GWAS hits (See also *Table 1* and *Figure 1—source data 1*) are highlighted in blue. Connecting lines indicate co-expression between the two transcripts.

The online version of this article includes the following source data for figure 1:

**Source data 1.** MMnet GWAS hits and their association with heel bone mineral density.

## MMnet genes have counter regulatory roles in the maintenance of adult bone mass and strength

To investigate whether individual MMnet genes regulate skeletal homeostasis, we studied all the MMnet genes for which the Wellcome Trust Sanger Institute, as part of the IMPC, had generated KO mice. Rapid-throughput phenotyping (*Bassett et al., 2012a*; *Estrada et al., 2012*; *Kemp et al., 2017*; *Kim, 2018*; *Medina-Gomez et al., 2018*; *Morris et al., 2019*) of samples from 16 week-old female mice with deletion of 12 MMnet genes (*Appl2*[-/-], *Atp8b2*[+/-], *Deptor*[-/-], *Dot1l*[+/-], *Egfr*[+/-], *Eml1*[-/-], *Eps15*[-/-], *Igsf8*[-/-], *Pik3cb*[-/-], *Rnf125*[-/-], *Slc1a4*[-/-], *Tgfb1i1*[-/-] mice) was performed using X-ray microradiography, micro-CT and biomechanical testing (*Figure 3A*). Six of the 12 MMnet KO mouse lines exhibited abnormal skeletal phenotypes, whereas *Dot1l*[+/-], *Egfr*[+/-], *Eps15*[-/-], *Rnf125*[-/-], *Slc1a4*[-/-] and *Tgfb1i1*[-/-] mice displayed no skeletal abnormalities (*Figure 3B*). *Igsf8*[-/-] mice had decreased femur length (*Figure 3—source data 1*; *Pik3cb*[-/-] (increased BV/TV and Tb.N), *Appl2*[-/-] (increased BV/TV) and *Atp8b2*[+/-] (increased BV/TV) mice had high bone mass; whereas *Deptor*[-/-] (decreased BV/TV and Tb.N) mice had low bone mass (*Figure 3C–D*, *Figure 3—source data 1*). Biomechanical testing demonstrated decreased bone strength in *Atp8b2*[+/-] (decreased femur stiffness), *Deptor*[-/-] (decreased vertebral stiffness) and *Eml1*[-/-] (decreased vertebral stiffness) mice (*Figure 3E–F*, *Figure 3—source data 1*).

## Human MMnet orthologues regulate osteoclast multinucleation and resorption

To investigate whether the divergent skeletal phenotypes identified in MMnet knockout mice correlate with osteoclast multinucleation and resorption in vitro, we determined the functional consequences of siRNA-mediated knockdown of 11 human orthologues of these MMnet genes in human osteoclasts (*TGFB1I1* was not expressed). The MMnet gene *DCSTAMP* was included as a positive control (*Figure 4A*, *Figure 4—figure supplement 1A*). For all 11 MMnet genes, siRNA-mediated knockdown resulted in a greater than 80% reduction in mRNA expression and did not affect cell viability (*Figure 4—figure supplement 1B–C*). The number of nuclei in each TRAP+ osteoclast was determined following siRNA knockdown (*Figure 4B–D*, *Figure 4—figure supplement 1D–E*). The functional consequences of in vitro siRNA knockdown correlated with the skeletal consequences of gene deletion in vivo. Knockdown of *DOTL1*, *EPS15* and *RNF125* had no effect on osteoclast multinucleation or resorption (*Figure 4B–D*) and the skeletal phenotype of the corresponding knockout mice was also unaffected (*Figure 3—source data 1*). Furthermore, knockdown of *APPL2* inhibited osteoclast resorption and knockdown of *PIK3CB* and *ATP8B2* inhibited both multinucleation and resorption (*Figure 4B–D*) whilst the corresponding knockout mice displayed increased bone mass (*Figure 3C–D*). By contrast, knockdown of *DEPTOR* resulted in a marked stimulation of both multinucleation and resorption (*Figure 4B–D*), and *DEPTOR* deficient mice had reduced bone mass and vertebral stiffness (*Figure 3C–F*). Overall, the effects of mRNA knockdown of MMnet genes on in vivo osteoclast multinucleation and resorption were concordant and strongly correlated ($R^2 = 0.76$; p<0.001) (*Figure 4E*). Moreover, the effect of knockdown of MMnet genes on osteoclast resorption

**Table 1.** MMnet is significantly enriched for variants that associate with different bone traits.

Single nucleotide polymorphisms (SNPs) significantly associated with the reported GWAS traits were interrogated in two datasets: the background dataset which includes all the eQTLs (posterior probability >80% see Materials and methods) and MMnet gene set. The enrichment was then tested over the background using a hypergeometric test. The significantly enriched traits are shown in bold.

| GWAS trait | Number of eQTLs genes (background) with at least one GWAS hit | Number of MMnet genes with at least one GWAS hit | Frequency (%) eQTLs genes (background) with at least one GWAS hit | Frequency (%) MMnet genes with at least one GWAS hit | P-value MMnet enrichment over background (hypergeometric test) |
|---|---|---|---|---|---|
| All GWAS combined | 299 | 41 | 20.3% | 27.7% | 0.0085 |
| Heel bone mineral density | 117 | 20 | 8.0% | 13.5% | 0.0045 |
| Osteoarthritis | 9 | 3 | 0.6% | 2.0% | 0.0083 |
| Psoriatic arthritis | 2 | 1 | 0.1% | 0.7% | 0.0101 |
| Body height | 165 | 24 | 11.2% | 16.2% | 0.0186 |
| Bone quantitative ultrasound measurement | 4 | 1 | 0.3% | 0.7% | 0.0527 |
| Bone density | 16 | 3 | 1.1% | 2.0% | 0.0686 |
| Hip bone mineral density | 5 | 1 | 0.3% | 0.7% | 0.0821 |
| Spine bone mineral density | 7 | 1 | 0.5% | 0.7% | 0.1509 |
| Bone fracture | 3 | 0 | 0.2% | 0.0% | 0.2727 |
| Ankylosing spondylitis | 4 | 0 | 0.3% | 0.0% | 0.3460 |
| Periodontitis | 4 | 0 | 0.3% | 0.0% | 0.3460 |
| Rheumatoid arthritis | 15 | 1 | 1.0% | 0.7% | 0.4550 |
| Adolescent idiopathic scoliosis | 60 | 5 | 4.1% | 3.4% | 0.5723 |
| Bone mineral content measurement | 1 | 1 | 0.1% | 0.7% | - |
| Lumbar disc degeneration | 1 | 1 | 0.1% | 0.7% | - |
| Osteitis deformans | 1 | 0 | 0.1% | 0.0% | - |

was also strongly and inversely correlated with trabecular bone mass (BV/TV) in knockout mice ($R^2$ = 0.65; p<0.01) (*Figure 4F*).

These findings demonstrate that the effects of MMnet gene deletion on the skeletal phenotype of knockout mice result, at least in part, from the direct consequences of gene deletion on osteoclast multinucleation and function, and that MMnet contains genes with important counter regulatory roles in the physiological control of adult bone mass and strength.

## MMnet regulates bone mass and strength via direct actions in the macrophage-osteoclast lineage

MMnet contains 190 genes, of which 178 are annotated, and among these genes there are known regulators of osteoclast multinucleation (*Dcstamp*, *Mmp9*, *Ctsk*) but the majority of MMnet genes require further investigation. To determine if MMnet genes regulate adult bone mass and strength by direct actions in osteoclasts we developed a prioritisation pipeline to identify a novel and

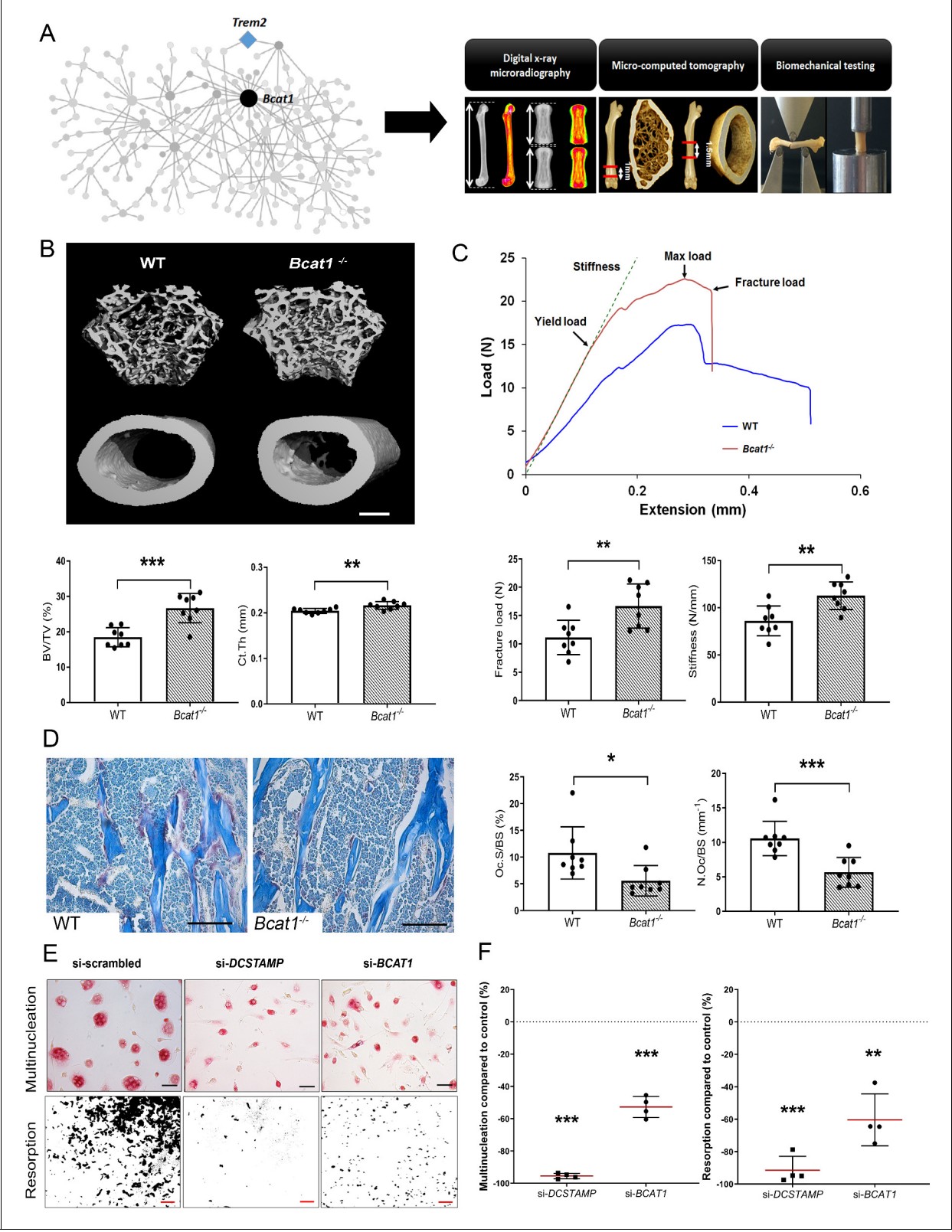

**Figure 2.** *Bcat1* deficiency results in high bone mass due to decreased osteoclast fusion. (**A**) Location of *Bcat1* as the MMnet hub (left panel). Skeletal phenotyping of *Bcat1*$^{-/-}$ mice using X-ray microradiography, micro computed tomography (micro-CT) and three point bending testing (right panel). (**B**) Micro-CT images of distal femur trabecular bone and mid-diaphysis cortical bone in wild-type (WT) and *Bcat1*$^{-/-}$ mice (upper panel) (scale bar = 200 µm). Trabecular bone volume/tissue volume (BV/TV) and cortical thickness (Ct.Th) are shown in the lower panel; (Mean ± SD; n = 8 per genotype,

*Figure 2 continued on next page*

*Figure 2 continued*

\*\*p<0.01, \*\*\*p<0.001 vs WT Student's t-test). (**C**) Representative load displacement curves from femur three-point bend testing (upper panel). Fracture load and stiffness are shown in the lower panel; (Mean ± SD; n = 8 mice per genotype, \*\*p<0.01 vs WT Student's t-test). (**D**) Representative tartrate-resistant acid phosphatase (TRAP) stained sections from the distal femur of WT and *Bcat1*-/- mice showing osteoclasts in red (left panel). Quantification of the osteoclast surface/bone surface (Oc.S/BS) and osteoclast number/bone surface (N.Oc/BS) (right panel) (scale bar = 100 μm); (Mean ± SD; n = 8 per genotype, \*p<0.05, \*\*\*p<0.001 vs WT Student's t-test). (**E**) Representative images of TRAP+ multinucleated osteoclasts (upper panel) (scale bar = 40 μm) and hydroxyapatite resorption (lower panel) (scale bar = 1 mm) 2 days after siRNA transfection. (**F**) Percentage inhibition of multinucleation compared to scrambled siRNA (left panel) and percentage inhibition of hydroxyapatite resorption compared to scrambled siRNA (right panel); (Mean ± SD from n = 4 donors, \*\*p<0.01, \*\*\*p<0.001 vs scrambled siRNA, one-sample-t test).

The online version of this article includes the following figure supplement(s) for figure 2:

**Figure supplement 1.** Deletion of *Bcat1* results in increased bone mass and strength due to impaired osteoclast multinucleation and function.

tractable MMnet gene to target for cell-specific gene deletion in the macrophage-osteoclast cell lineage. To identify potential novel and tractable candidates we applied a bespoke in silico filtering to the 178 annotated MMnet genes. The following criteria were used as a stepwise prioritisation filter (*Figure 5—source data 1*): (i) conserved expression in human osteoclasts; (ii) conserved expression in mouse macrophages; (iii) unknown function in osteoclastogenesis (Pubmed, accessed December 2017); (iv) availability of the mouse model (http://www.informatics.jax.org/); (v) strength of the MMnet eQTL ($R^2$ >1) (*Kang et al., 2014*; (vi) Membrane receptors with pharmacologically tractable signalling pathways (*Figure 5—figure supplement 1A*, *Figure 5—source data 1*). This prioritisation process identified five candidate MMnet genes (*FGR*, *LAT2*, *MYOF*, *FCGRT*, and *SLC40A1*) (*Figure 5—figure supplement 1A–B*). To identify the most robust regulator of osteoclast multinucleation we determined the consequences of siRNA-mediated knockdown of these five genes in human osteoclasts. Knockdown of all five MMnet genes resulted in greater than 40% inhibition of multinucleation (*Figure 5—figure supplement 1C*). Most strikingly, knockdown of *SLC40A1* (encoding the mammalian iron transporter, ferroportin) resulted in more than 80% inhibition of multinucleation, an effect comparable with that of the master regulator *DCSTAMP* (*Figure 5—figure supplement 1C–D*).

Consequently, SLC40A1 was prioritised for further investigation and we generated a myeloid lineage-specific knockout of *Slc40a1* by crossing *Slc40a1*flox/flox mice with LysM-cre mice. Femurs from 16 week-old male and female mutant Slc40a1 conditionnal KO (cKO) and Control mice were analysed by X-ray-microradiography, microCT and 3-point bend testing (n = 8 per sex per genotype). Slc40a1 cKO mice had greatly increased bone mass characterised by increased BMC (*Figure 5B–C*), cortical thickness (Ct.Th), trabecular BV/TV, Tb.N and Tb.Th compared to littermate controls (*Figure 5C–D*, *Figure 5—figure supplement 2A*). This high bone mass phenotype was associated with markedly increased bone strength characterised by increased yield, maximum and fracture loads (*Figure 5C–D*, *Figure 5—figure supplement 2A*). Consistent with this, histomorphometic analysis demonstrated decreased osteoclast surfaces (Oc.S/BS) in Slc40a1 cKO mice (*Figure 5E*) and Slc40a1 cKO mice had increased amounts of retained cartilage within trabecular bone, consistent with impaired osteoclastic bone resorption and abnormal bone remodelling (*Figure 5—figure supplement 2 B, C*). siRNA-mediated knockdown of *SLC40A1* expression, in human osteoclasts in vitro, inhibited osteoclast resorption by more than 85% (*Figure 5F–G*). Taken together, these data demonstrate that the MMnet gene, *Slc40a1*, acts as an important determinant of adult bone mass and strength by directly regulating osteoclast multinucleation and function.

## Discussion

In this study, we took advantage of an unique *trans*-eQTL network (MMnet) in multinucleating rat primary macrophages that is enriched for osteoclast genes. The parental rat strains used to generate MMnet (WKY and LEW) had opposing skeletal phenotypes and rate of osteoclast multinucleation. Spontaneous fusion of WKY macrophages in vitro was associated with decreased bone length, mass, and strength in vivo, suggesting that genetic determinants of osteoclast multinucleation directly affect bone development and maintenance. These results implicate MMnet in the physiological regulation of adult bone mass and strength.

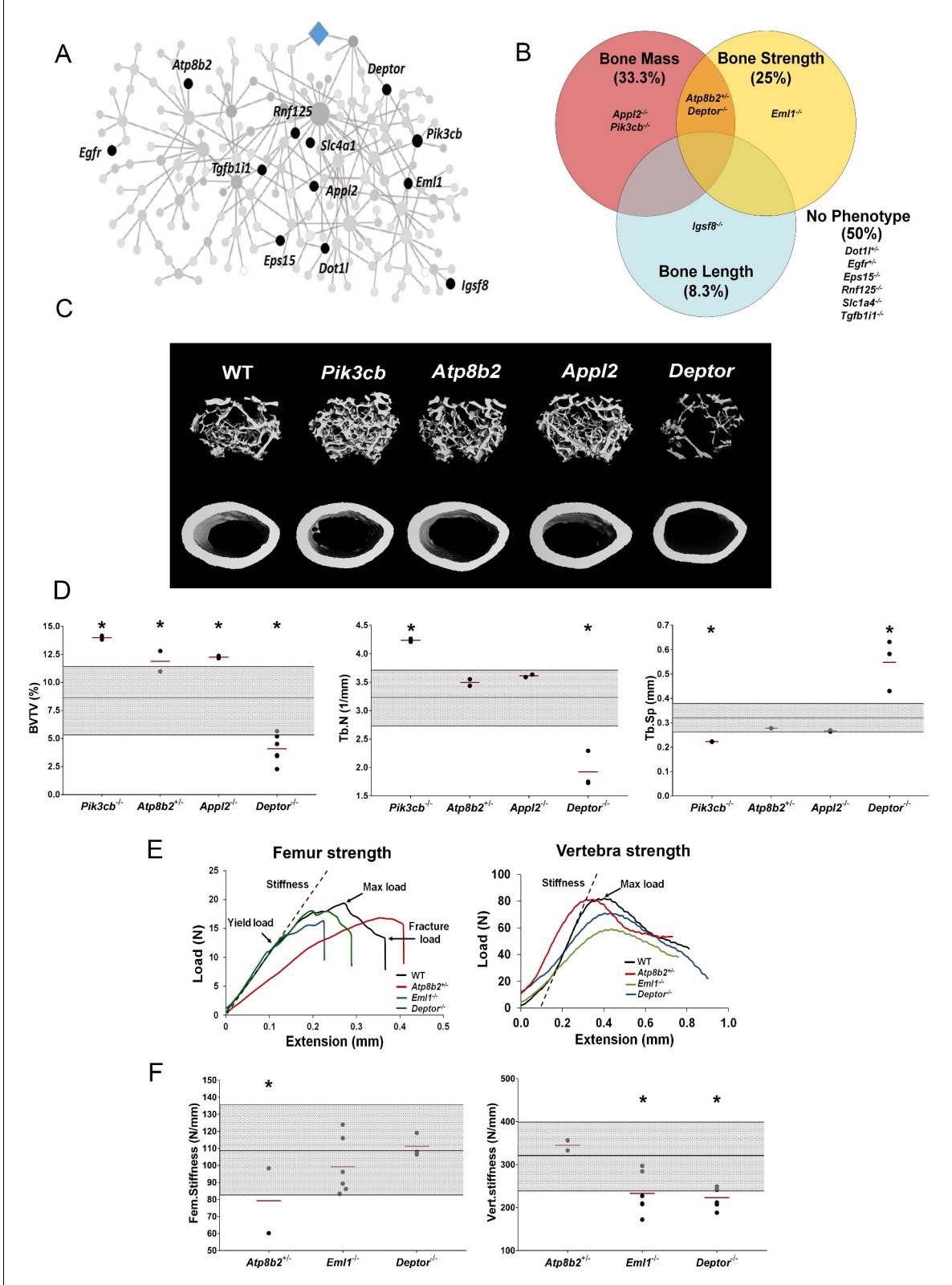

**Figure 3.** MMnet regulates bone mass. (**A**) The location of the 12 MMnet genes for which knockout mice were available are highlighted in the network. (**B**) Venn diagram showing mutant strains with outlier phenotypes for bone mass, bone strength and bone length. (**C**) Representative micro-CT images of distal femur trabecular bone and mid diaphyseal cortical bone from WT and mutant mice (scale bar = 200 µm). (**D**) Graphs show trabecular bone volume (BV/TV), trabecular number (Tb.N) and trabecular separation (Tb.Sp). Grey box represents WT reference mean + / - 2 SD (n = 320, 16 week old,

*Figure 3 continued on next page*

*Figure 3 continued*

female C57BL/6 WT mice). (**E**) Representative load displacement curves for femur three-point bend testing and vertebral compression. (**F**) Femur stiffness (Fem.stiffness) and vertebral stiffness (vert.stiffness). Data from individual mice are shown as black dots and mean value by a red line (n = 2 or six per genotype). Parameters outside the WT reference range are indicated by an asterisk *.

The online version of this article includes the following source data for figure 3:

**Source data 1.** Results of high-throughput MMnet mouse knockout skeletal phenotyping.

Next, we determined whether MMnet is enriched for GWAS variants associated with skeletal traits and showed enrichment for heel bone mineral density, body height, osteoarthritis and psoriatic arthritis. These results confirm conservation of MMnet in humans and demonstrate associations between MMnet and osteoclast multinucleation and bone mass. Furthermore, the association of MMnet with osteoarthritis suggests involvement of osteoclasts in articular cartilage degradation, an area that is gaining increasing attention (*Goldring and Goldring, 2016*; *Löfvall et al., 2018*).

To validate the function of MMnet in vivo, we first investigated the role of the hub gene, *Bcat1*, in knockout mice and demonstrated important new roles for BCAT1 in osteoclast formation and function, and the physiological maintenance of adult bone mass and strength. These results demonstrate that MMnet has a critical homeostatic role in bone that is conserved in rodents and humans. Furthermore, they specifically implicate branched chain amino acid (BCAA) metabolism in osteoclast-mediated bone homeostasis and are consistent with our recent observation that osteoclast numbers and severity of collagen-induced arthritis are also reduced following pharmacological inhibition of BCAT1 in mice (*Papathanassiu et al., 2017*). BCAT1 activity controls intracellular leucine, a known activator of mammalian target of rapamycin complex 1 (mTORC1) signalling (*Ananieva et al., 2014*), suggesting that MMnet may directly regulate mTORC1-dependent bone homeostasis and inflammation via the hub gene *Bcat1*.

To investigate further the homeostatic roles of MMnet in the maintenance of adult bone mass and strength, we analysed knockout mice with deletion of 12 unselected MMnet genes and demonstrated significant outlier phenotypes in six of these lines, which include both known *Pik3cb* (*Győri et al., 2014*; *Hall et al., 1995*; *Lee et al., 2002*) and unknown (*Atp8b2, Igsf8, Eml1, Appl2, and Deptor*) regulators of skeletal homeostasis. Deletion of *Pik3cb, Appl2* and *Atp8b2* resulted in increased bone mass whereas deletion of *Deptor* resulted in decreased bone mass and strength, suggesting MMnet represents a conserved and complex homeostatic counter-regulatory network that serves to optimise bone structure and strength. Consistent with this, deletion of MMnet's strongest *trans*-eQTL *Kcnn4* results in increased bone mass (*Kang et al., 2014*) as did deletion of *Dcstamp,* the master regulator of membrane fusion (*Yagi et al., 2005*). Nevertheless, although (i) BCAT1, PIK3CB, ATP8B2, and APPL2 knockdown in vitro resulted in impaired osteoclast multinucleation and resorption in humans, and (ii) *Bcat1, Pik3cb, Atp8b2*, and *Appl2* deletion in vivo resulted in increased trabecular bone mass, only *Bcat1* deficient mice exhibited increased cortical thickness and bone strength. These findings may refect that (i) *Bcat1* deletion has the greatest effect on osteoclast multinucleation and function in vivo, (ii) *Atp8b2* mice were heterozygotes, (iii) trabecular bone is more sensitive than cortical bone to abnormalities of osteoclastic resorption, and (iv) cortical bone parameters are the major determinants of bone strength. However, since these genetically modified mice have global gene deletions, the variation in skeletal phenotypes may also result from the consequences of gene deletion in non-monocyte/macrophage lineages.

Overall, these studies reveal a common molecular pathway within MMnet that regulates osteoclast multinucleation and function as well as bone mass. Thus, at least four MMnet genes (*Trem2, Bcat1, Pik3cb* and *Deptor*), that include the network master regulator (*Trem2*) and network hub (*Bcat1*), act within the Trem2-Pi3K-mTORC1 pathway (*Ananieva et al., 2014*; *Mossmann et al., 2018*; *Peterson et al., 2009*; *Ulland et al., 2017*). mTORC1 protein kinase comprises a complex of proteins that includes DEPTOR (*Peterson et al., 2009*), a regulator of cell growth and metabolism that is essential for osteoclast formation (*Indo et al., 2013*). While *Trem2* (*Ulland et al., 2017*), *Bcat1* (*Ananieva et al., 2014*) and *Pik3cb* (*Mossmann et al., 2018*) are positive regulators of mTORC1, *Deptor* inhibits mTORC1 activity (*Peterson et al., 2009*). Significantly, the divergent effects of these genes on activity of the mTORC1 signalling pathway correlate precisely with the effects of gene knockdown or deletion on osteoclast multinucleation and function in vitro and on skeletal phenotype in vivo. Thus, knockdown or deletion of *Bcat1* and *Pik3cb* results in decreased

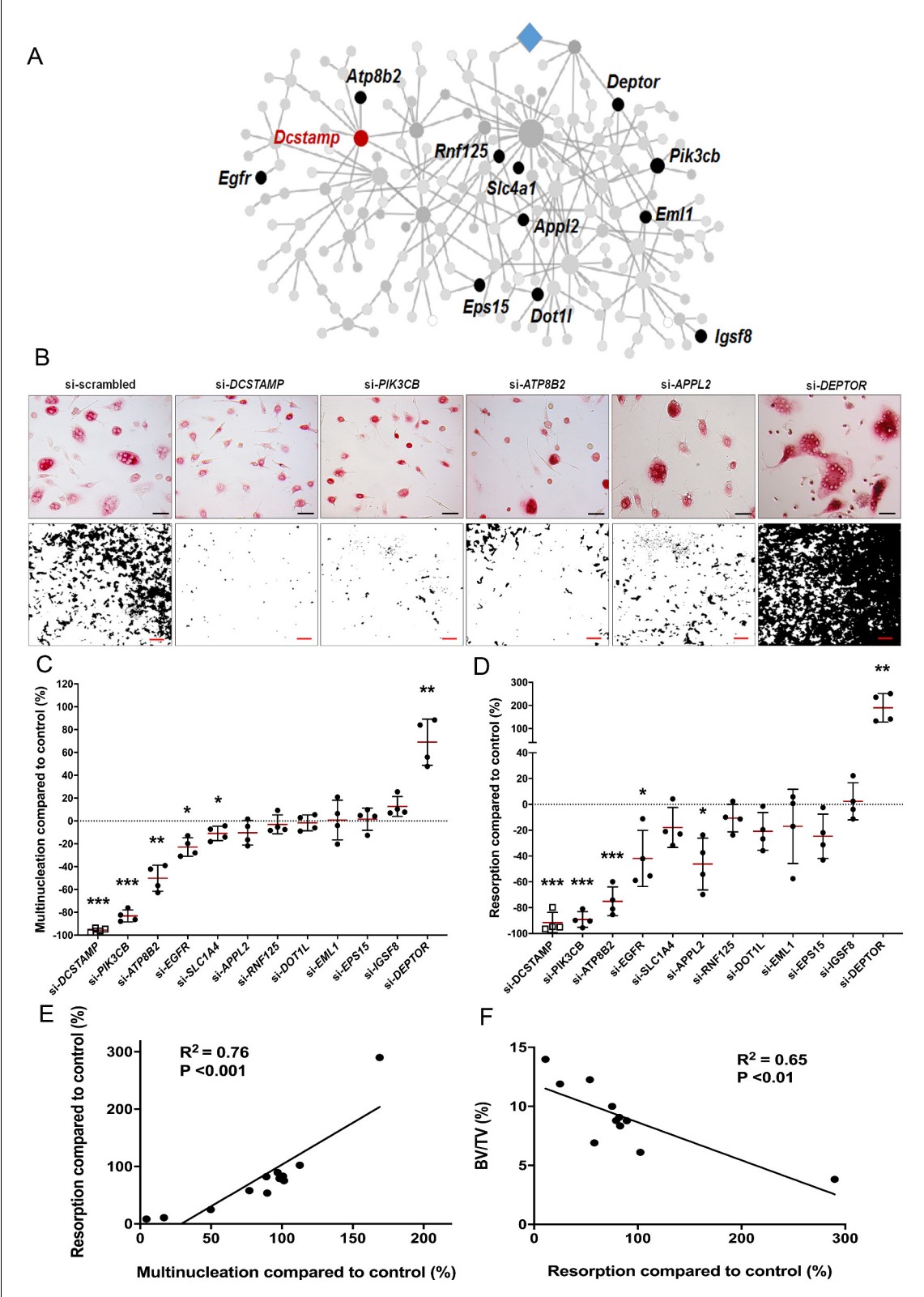

**Figure 4.** MMnet regulates human osteoclast multinucleation and resorption in vitro. (**A**) The location of the 11 MMnet genes for which knockout mice were available, and for which there are human orthologues. The master regulator of macrophage fusion *Dcstamp* is highlighted in red. (**B**) Representative images of TRAP+ osteoclasts (Top panel) (scale bar = 40 μm) and hydroxyapatite resorption (Bottom panel) (scale bar = 1 mm) following siRNA knockdown for *DCSTAMP*, *PIK3CB*, *ATP8B2*, *APPL2* and *DEPTOR*. (**C**) Graphs show the frequency of multinucleated cells (>4 nuclei) following

*Figure 4 continued on next page*

*Figure 4 continued*

siRNA knockdown as a percentage of the scramble siRNA control; n = 4 donors. (D) Hydroxyapatite resorption following siRNA knockdown as a percentage of the scramble siRNA control; n = 4 donors. (For (C) and (D): Mean ± SD from n = 4 donors, *p<0.05, **p<0.01, ***p<0.001 vs scrambled siRNA, one-sample-t test). (E) Pearson correlation between human osteoclast multinucleation and resorptive activity following siRNA knockdown of 12 MMnet genes compared to control ($R^2$ = 0.76; p<0.001) (F) Pearson correlation between human osteoclast resorption activity following si-RNA knockdown of 11 MMnet genes compared to control and BV/TV in the corresponding 11 MMnet knockout mouse strains ($R^2$ = 0.65; p<0.01).
The online version of this article includes the following figure supplement(s) for figure 4:

**Figure supplement 1.** siRNA mediated knockdown of MMnet genes in human osteoclasts.

osteoclast multinucleation and resorption, and increased bone mass, whereas knockdown or deletion of *Deptor* results in opposite cellular and in vivo phenotypes. Together, these data suggest a pivotal and integrated counter-regulatory role for the Trem2-Pi3K-mTORC1 pathway in osteoclast multinucleation and function, and the homeostatic control of bone mass, and may thus provide novel therapeutic targets for the inhibition of bone loss.

Overall, these studies demonstrate that MMnet regulates bone mass and strength via important effects on osteoclast multinucleation and function. Nevertheless, it is important to recognise that primary actions of MMnet genes in osteoclasts may secondarily regulate osteoblastic bone formation. Alternatively, MMnet genes may have direct cell autonomous effects in the osteoblast lineage. Thus, direct and indirect effects of MMnet genes in osteoblast may influence the skeletal phenotypes observed in mutant mice. The role of osteoblasts has not been investigated here, and this represents an important limitation of the study.

Finally, we investigated the role of the iron transporter ferroportin, encoded by the MMnet gene *Slc40a1*, in the osteoclast lineage. Ferroportin was identified as a powerful inhibitor of human osteoclast multinucleation and function in vitro, with a potency equivalent to *Dcstamp* (*Figure 5* and *Figure 4—figure supplement 1*). Generation of mice with conditional deletion of *Slc40a1* in the macrophage-osteoclast lineage using *LysMCre* mice was consistent with in vitro multinucleation and resulted in a phenotype of increased bone mass and strength (*Figure 5* and *Figure 5—figure supplement 2*). Thus, *Slc40a1* regulates bone structure and strength via direct actions in the macrophage-osteoclast lineage, further supporting the conclusion that MMnet plays a key role in the physiological regulation of bone mass by osteoclasts.

Nevertheless, Wang and colleagues (*Wang et al., 2018*) recently reported that deletion of *Slc40a1* in the myeloid lineage using *LysMCre* mice causes decreased bone mass and mineralisation in female, but not male, mice. Wang et al also reported that deletion of *Slc40a1* in terminally differentiated osteoclasts using *CtskCre* mice resulted in no skeletal abnormalities in either sex. The discordance between our findings and those reported by Wang et al may be due to differences in the age at which mice were phenotyped and the strain of *Slc40a1* floxed mice investigated (*C57BL/6N-Slc40a1$^{tm1c(EUCOMM)Hmgu/H}$* in this study compared to *129S-Slc40a1$^{tm2Nca/J}$* by Wang and colleagues). Although it is clear that ferroportin acts directly in the macrophage-osteoclast lineage, further longitudinal studies will be required to define its temporal effects on bone mass and strength, whilst studies with the different floxed strains may be required to determine whether modifier genes influence the skeletal response to deletion of *Slc40a1*.

Systems genetics studies use naturally occurring genetic variation to identify gene networks that are associated with the trait of interest in disease-relevant cells (*Baliga et al., 2017*). Systems genetics approaches in osteoblasts and osteoclasts have been previously carried out to identify genes that are implicated in skeletal homeostasis (*Calabrese et al., 2012*; *Calabrese et al., 2017*; *Farber et al., 2011*; *Mesner et al., 2019*; *Mesner et al., 2014*). Here, we have used systems genetics approaches to demonstrate that a complex *trans*-eQTL network (MMnet) facilitates homeostatic control of bone mass via its effects on osteoclast fusion and function that are mediated in large part by the mTOR pathway. This physiologically important network is conserved in rats, mice and humans and its identification will lead the way to an understanding of the gene-gene interactions and network-based therapeutic approaches that may be used in osteoclast-mediated inflammatory disease or to prevent bone loss.

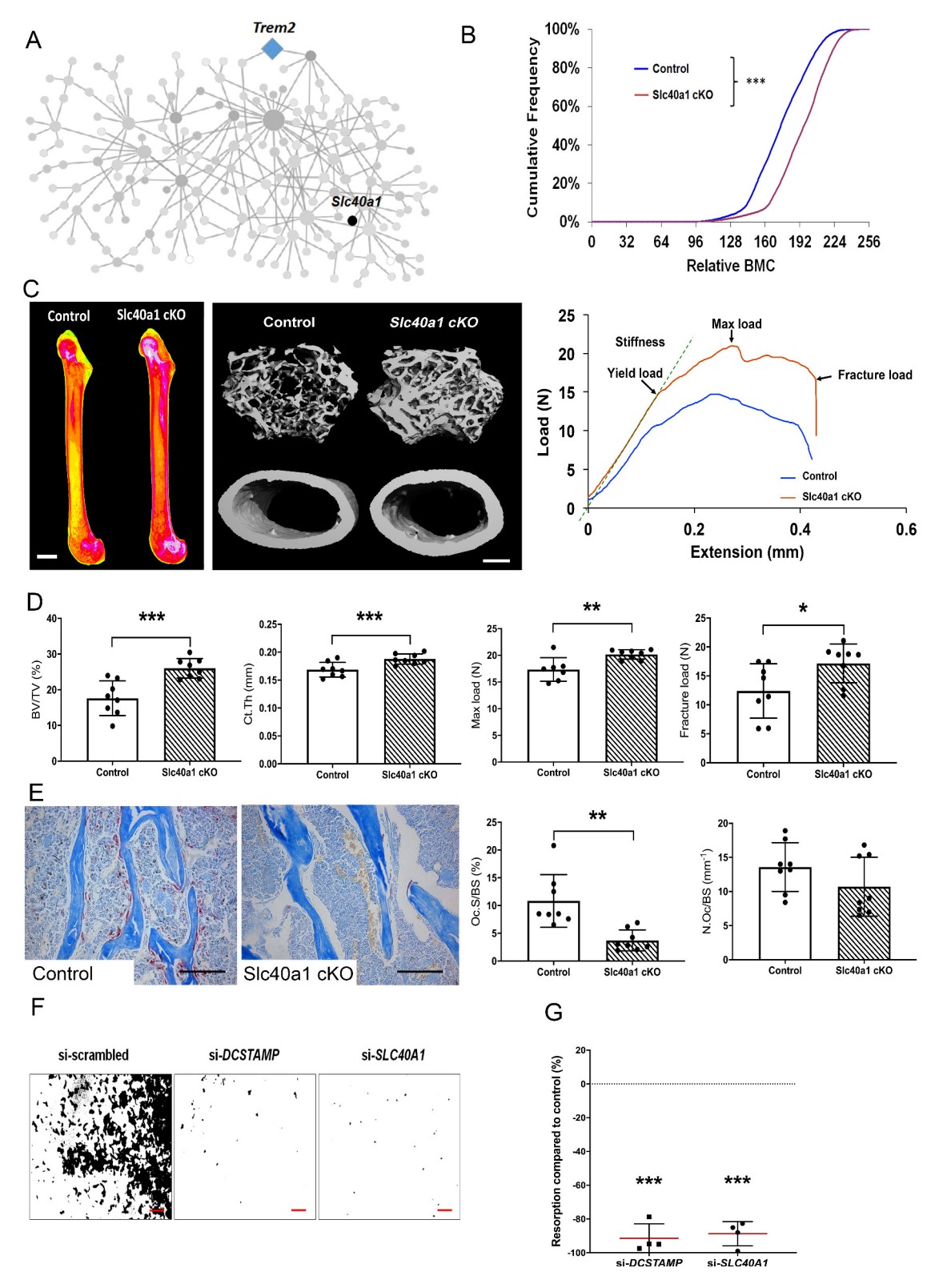

**Figure 5.** Myeloid-specific deletion of *Ferroportin* results in increased bone mass and strength due to impaired osteoclast multinucleation and function. (A) Location of *Slc40a1* in MMnet. (B) Cumulative frequency histogram of relative BMC; n = 8 per genotype. ***p<0.001 vs. WT by Kolmogorov–Smirnov test. (C) Representative X-ray microradiography images of femurs from male Control and Slc40a1 cKO mice. Pseudocoloured images indicating low BMC (blue and green) and high BMC (red and pink) are shown in the left panel (scale bar = 1 mm). Micro-CT images of distal femur trabecular bone

*Figure 5 continued on next page*

**Figure 5 continued**
and mid-diaphysis cortical bone (middle panel) (scale bar = 200 µm). Representative load displacement curves from femur three-point bend testing (right panel). (D) Trabecular bone volume (BV/TV) and cortical thickness (Ct.Th), femur maximum load and fracture load. (E) Representative TRAP-stained femur sections from Control and Slc40a1 cKO mice showing osteoclasts stained in red (left panel). Osteoclast surface (Oc.S/BS) and osteoclast number (N.Oc/BS) (right panel) (scale bar = 100 µm); (For (D) and (E): Mean ± SD; n = 8 per genotype, *p<0.05, **p<0.01, ***p<0.001 vs Control Student's t-test). (F) Representative images of hydroxyapatite resorption (scale bar = 1 mm) following *DCSTAMP* and *SLC40A1* si-RNA knockdown. (G) Inhibition of osteoclast hydroxyapatite resorption following *DCSTAMP* and *SLC40A1* siRNA knockdown; (Mean ± SD from n = 4 donors, ***p<0.001 vs scrambled siRNA, one-sample-t test).

The online version of this article includes the following source data and figure supplement(s) for figure 5:

**Source data 1.** Prioritisation pipeline to identify a novel and tractable MMnet gene.
**Figure supplement 1.** An In silico filtering strategy prioritises *Slc40a1* (ferroportin) for myeloid-targeted gene deletion.
**Figure supplement 2.** Myeloid-specific deletion of *Ferroportin* results in increased bone mass and strength due to impaired osteoclast multinucleation and function.

# Materials and methods

## Animals

14 week-old male Wistar-Kyoto (WKY/NCrl) and Lewis (LEW/Crl) rats were purchased from Charles River. *Bcat1* knockout (*Bcat1⁻ᐟ⁻*) and wild-type (WT) littermate mice were previously described (*Ananieva et al., 2014*) and maintained according to protocols approved by the University of Virginia Animal Care and Use Committee. To rederive *Slc40a^flox/flox^* mice, sperm from conditional-ready heterozygous *Slc40a1*-floxed mice (C57BL/6N-Slc40a1^tm1c(EUCOMM)Hmgu/H^) was obtained from MRC Harwell and used for in vitro fertilisation of C57BL/6N female mice. LysM-Cre mice (*Lyz2^tm1(cre)Ifo^*) were obtained from The Jackson Laboratory. *Slc40a1^flox/flox^* and LysM-Cre mice were crossed to generate Slc40a1 mice in which *Slc40a1* is deleted in cells of the monocyte-macrophage-osteoclast lineage (cKO). *Slc40a1^flox/flox^* (called Control) littermates were used as controls in all experiments. All animals were housed in standard caging on a 12-hr-light cycle and provided with free access to chow and water.

Tissues were collected and placed in either 70% ethanol or 10% neutral buffered formalin for 24 hr prior to storage in 70% ethanol. Samples were randomly assigned to batches for rapid-throughput analysis. All studies were performed in accordance to the U.K. Animal (Scientific Procedures) Act 1986, the ARRIVE guidelines, the EU Directive 2010/63/EU for animal experiments and practices prescribed by the National Institutes of Health in the United States.

The International Mouse Phenotyping Consortium (IMPC: http://www.mousephenotype.org) and the International Knockout Mouse Consortium (IKMC) are generating null alleles for all protein-coding genes in mice on a C57BL/6 genetic background (*Collins et al., 2007*). The Origins of Bone and Cartilage Disease Programme (OBCD) is undertaking a validated rapid-throughput multiparameter skeletal phenotype screen (*Bassett et al., 2012a*; *Freudenthal et al., 2016*) of mutant mouse lines generated by the Wellcome Trust Sanger Institute as part of the IKMC and IMPC effort.

## Digital X-ray microradiography

Femurs and caudal vertebrae 6 and 7 from 16 week-old mice and 14 week-old rats were fixed in 70% ethanol. Soft tissue was removed and digital X-ray images were recorded at 10 µm pixel resolution using a Faxitron MX20 variable kV point projection X-ray source and digital image system (Qados, Cross Technologies plc, UK) operating at 26 kV and 5x magnification. Bone mineral content (BMC) was determined relative to steel, aluminium and polyester standards as described (*Bassett et al., 2012b*). Images were calibrated with a digital micrometer and long bone and vertebra lengths were determined (*Bassett et al., 2012a*; *Bassett et al., 2012b*).

## Micro-computerised tomography (micro-CT)

Mouse and rat femurs were analysed by micro-CT (Scanco uCT50, 70 kV, 200 µA, 0.5 mm aluminium filter) as described (*Bassett et al., 2012a*). Measurements included cortical bone parameters (cortical thickness (Ct.Th), cortical bone mineral density (Ct.BMD), internal diameter (Int.Dia) and bone area (Ct.Ar) at 10 µm³ voxel resolution in a 1.5 mm² region centred on the mid-shaft region 56% of the way along the length of the femur distal to the femoral head, and trabecular parameters (bone

volume (BV/TV), trabecular number (Tb.N), thickness (Tb.Th), spacing (Tb.Sp), structure model index (SMI) and trabecular BMD (Tb.BMD)) at 5 µm$^3$ voxel resolution in a 1 mm$^2$ region beginning 100 µm proximal to the distal growth plate. Parameters were determined using Scanco analysis software.

## Destructive 3-point bend testing

Destructive 3-point bend tests were performed on femurs and compression tests on caudal vertebrae 6 and 7 using an Instron 5543 load frame and load cell (100N for mouse femurs and 500N for mouse caudal vertebrae and rat bones) (Instron Limited, High Wycombe, Buckinghamshire, UK) as described (Bassett et al., 2012a). Bones were positioned horizontally on custom supports and load was applied perpendicular to the mid-diaphysis with a constant rate of displacement of 0.03 mm/s until fracture. Vertebrae were bonded in vertical alignment to a custom anvil support using cyanoacrylate glue and load was applied vertically at a constant rate of displacement of 0.03 mm/s and a sample rate of 20 Hz. Yield load, maximum load, fracture load, stiffness and toughness (Energy dissipated prior to fracture) were determined from femur load displacement curves and yield load, maximum load and stiffness from caudal vertebrae load displacement curves.

## Osteoclast histomorphometry

Osteoclast numbers were determined according to the American Society for Bone and Mineral Research system (Dempster et al., 2013) in paraffin sections from decalcified mouse femurs stained for tartrate resistant acid phosphatase (TRAP) activity, counterstained with aniline blue and imaged using a Leica DM LB2 microscope and DFC320 digital camera (Bassett et al., 2014). A montage of nine overlapping fields covering an area of 1 mm$^2$ located 0.2 mm below the growth plate was constructed for each bone. BV/TV was measured, and osteoclast numbers and surface were determined in trabecular bone normalised to total bone surface (Bassett et al., 2014).

## Alcian blue and Van Gieson staining of retained cartilage

Paraffin sections from decalcified mouse femurs were stained with Alcian blue van Gieson. Sections were imaged using a Leica DM LB2 microscope at a resolution of 920 pixels/mm. The trabecular region of interest (ROI) was defined as a 1 mm$^2$ region beginning 100 µm proximal to the distal growth plate. The cortical ROI was defined as two 1.0 × 0.3 mm regions located 100 µm proximal to the distal growth plate each containing one femoral cortex. Blinded samples were analysed to determine bone area (B.Ar) and cartilage area (Cartilage.Ar) for each ROI using the Threshold and Colour Threshold functions of ImageJ. Cartilage area as a proportion of bone area (Cartilage.Ar/B.Ar) was then determined for both trabecular and cortical bone.

## Cell culture, transfection and assessment of cell multinucleation

Human monocyte-derived macrophages were separated from healthy donor buffy coats by centrifugation through a Histopaque 1077 (Sigma) gradient and adhesion purification. Following Histopaque separation, peripheral blood mononuclear cells were re-suspended in RPMI (Life Technologies) and monocytes purified by adherence for 1 hr at 37˚C, 5% CO$_2$ in twelve-well plates. The monolayer was washed three times with HBSS to remove non-adherent cells and monocytes were differentiated for 3 days in RPMI medium containing 20 ng/ml M-CSF and 20 ng/ml recombinant human RANKL (PeproTech, UK). After 3 days of culture, mononucleated TRAP positive osteoclasts were obtained and transfected using Dharmafect 1 (Dharmacon) diluted in OPTIMEM medium (1:50 Invitrogen). siGENOME SMARTpools (100 nM, Dharmacon SMART pool) targeting human *APPL2*, *ATP8B2*, *BCAT1*, *DEPTOR*, *DOT1L*, *EGFR*, *EML1*, *EPS15*, *IGSF8*, *PIK3CB*, *RNF125*, *SLC1A4* and *SLC40A1* were used for RNAi and a non-targeting siRNA pool was used as the scrambled control siRNA. Primer sequence information is available in Supplementary Method 1. At the end of the culture (Day 5), osteoclasts were fixed in formalin, stained for TRAP and imaged. Transfection experiments were performed with four donors per si-RNA and three technical replicates in 12-well plates. Following the transfection, the number of nuclei in each TRAP-positive cell was counted to obtain the percentage of cells with 1, 2, three and >4 nuclei, relative to numbers of nuclei in cells transfected with scrambled control siRNA.

Bone marrow-derived osteoclasts were isolated from 12 week old LEW and WKY rats by flushing bone marrow cells in DMEM (Thermo Fisher) supplemented with 25 mM HEPES buffer (Sigma), 25%

fetal bovine serum (Labtech), penicillin/streptomycin (100 units/ml; Thermo Fisher). Bone marrow cells were differentiated into osteoclasts in DMEM containing 20 ng/ml M-CSF and 20 ng/ml recombinant rat RANKL (PeproTech, UK) for 6 days in Petri dishes (Nunc). At the end of the culture, cells were fixed in formalin and stained with Giemsa to visualise nuclei and count osteoclasts.

## Quantitative RT-PCR

For quantitative RT-PCR (RT-qPCR), total RNA was extracted from human and rat osteoclasts using the TRIzol reagent (Invitrogen, Carlsbad, CA) according to the manufacturer's instructions. Complementary DNA (cDNA) was synthesised using iScript cDNA Synthesis Kit (Bio-Rad, Hercules, CA). A total of 10 ng cDNA for each sample was used and all RT-qPCR reactions were performed on a ViaA 7 Real-Time PCR System (Life Technologies, Carlsbad, CA) using Brilliant II SYBR Green QPCR Master Mix (Agilent, Santa Clara, CA). Results were analysed by the comparative Ct method using ViiA 7 RUO software, and each sample was normalised relative to *HPRT* expression.

## In vitro resorption

In vitro resorption activity of human osteoclasts was measured on Osteo Assay Surface 96 well plates (hydroxyapatatite surfaces) (Corning). siRNA transfected cells were incubated with cell dissociation buffer (Sigma) and $10^5$ cells/well were seeded in Osteo Assay Surface Plates. After 2 days of culture with 20 ng/ml M-CSF and 20 ng/ml recombinant human RANKL (PeproTech, UK), the wells were rinsed twice with PBS and incubated with 10% bleach solution for 30 min at room temperature. The wells were then washed twice with PBS and allowed to dry at room temperature. Individual resorption pits were imaged by light microscopy. Images were inverted and processed using Photoshop to yield high-contrast images and show the resorbed areas in black against a white background. Binary images of each individual well were then subjected to automated analysis (ImageJ), using constant 'threshold' and 'minimum particle' levels, to determine the number and surface area of resorbed pits.

## Statistical analysis

Data are presented as mean ± standard deviations (SD) and analysed using GraphPad Prism software (version 7.02; GraphPad). Normally distributed data were analysed by two tailed Student's t test. Relationships between micro-CT parameters and in vitro osteoclastogenesis were determined by Pearson correlation. Frequency distributions of bone mineral densities obtained by x-ray microradiography were compared using the Kolmogorov-Smirnov test. Differences in percentage of control following si-RNA knockdown in primary osteoclasts were tested for significance using a one-sample-t test.

IMPC mutant lines were compared to C57BL6/N strain-specific reference ranges established for all parameters using 16 week-old female wild-type mice (n = 320) obtained from control cohorts. Strains in which a structural or functional parameter was ±2.0 SD from the C57BL6/N reference mean were considered as outliers.

The hypergeometric test was used to determine whether the MMnet network (n = 190 rat genes, of which 178 genes had a unique human orthologue) was enriched for genes with a GWAS signal (p<$10^{-6}$) for traits related to bone and height (*Table 1*) in the NHGRI-EBI Catalog of published GWAS (https://www.ebi.ac.uk/gwas/). Over-representation of GWAS gene signals in MMnet was assessed in comparison with the whole set of eQTLs (n = 1527 genes, of which 1448 genes had a unique human orthologue) that were previously mapped in multinucleating macrophages (*Kang et al., 2014*). Biomart was used to map rat genes to the human orthologues (one-to-one orthology) using Ensembl rel. 98.

## Acknowledgements

We thank Mahrokh Nodani for technical assistance. We thank members of Sanger Mouse Pipelines (Mouse Informatics, Molecular Technologies, Genome Engineering Technologies, Mouse Production Team, Mouse Phenotyping) and the Research Support Facility for provision and management of mice. We thank Susan Hutson for providing the *Bcat1*$^{-/-}$ mice (KSP is supported by NIH/NCI U01CA224293). This work was supported by the Medical Research Council 'Control of Macrophage Multinucleation in Health and Disease' (MR/N01121X/1 to JB, GRW, JHDB), a Wellcome Trust

Strategic Award (Grant Number 101123 to GRW and JHDB) and National Institutes of Health/
National Cancer Institute (NIH/NCI U01CA224293).

## Additional information

### Funding

| Funder | Grant reference number | Author |
|---|---|---|
| Medical Research Council | MR/N01121X/1 | JH Duncan Bassett<br>Graham R Williams<br>Jacques Behmoaras |
| Wellcome | 101123 | JH Duncan Bassett<br>Graham R Williams |
| National Cancer Institute | U01CA224293 | Kwon-Sik Park |

The funders had no role in study design, data collection and interpretation, or the
decision to submit the work for publication.

### Author contributions

Marie Pereira, Conceptualization, Formal analysis, Validation, Investigation, Visualization, Methodology, Writing - original draft, Project administration, Writing - review and editing; Jeong-Hun Ko, Hayley Protheroe, Investigation; John Logan, Investigation, Visualization, Writing - review and editing; Kee-Beom Kim, Resources, Investigation; Amelia Li Min Tan, Data curation, Formal analysis; Peter I Croucher, Resources; Kwon-Sik Park, Resources, Writing - review and editing; Maxime Rotival, Methodology, Writing - review and editing; Enrico Petretto, Formal analysis, Validation, Writing - review and editing; JH Duncan Bassett, Graham R Williams, Conceptualization, Resources, Supervision, Funding acquisition, Validation, Visualization, Methodology, Project administration, Writing - review and editing; Jacques Behmoaras, Conceptualization, Resources, Supervision, Funding acquisition, Validation, Visualization, Methodology, Writing - original draft, Project administration, Writing - review and editing

### Author ORCIDs

Marie Pereira (iD) https://orcid.org/0000-0003-0711-3385
JH Duncan Bassett (iD) https://orcid.org/0000-0003-0817-0082
Graham R Williams (iD) https://orcid.org/0000-0002-8555-8219

### Ethics

Animal experimentation: All studies were performed in accordance to the U.K. Animal (Scientific Procedures) Act 1986, the ARRIVE guidelines, the EU Directive 2010/63/EU for animal experiments and practices prescribed by the National Institutes of Health in the United States.

### Decision letter and Author response

Decision letter https://doi.org/10.7554/eLife.55549.sa1
Author response https://doi.org/10.7554/eLife.55549.sa2

## Additional files

### Supplementary files

• Supplementary file 1. List of primers used.

• Transparent reporting form

### Data availability

All data generated or analysed during this study are included in the manuscript and supporting files.

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
