## [Decision Letter]

**Decision letter after peer review:**

Thank you for submitting your article "A trans-eQTL network regulates osteoclast multinucleation and bone mass" for consideration by *eLife*. Your article has been reviewed by Clifford Rosen as the Senior Editor, a Reviewing Editor, and three reviewers. The following individuals involved in review of your submission have agreed to reveal their identity: Natalie Sims (Reviewer #1); Nathan J Pavlos (Reviewer #2); Steven Munger (Reviewer #3).

The reviewers have discussed the reviews with one another and the Reviewing Editor has drafted this decision to help you prepare a revised submission.

Summary:

This manuscript describes a systems genetics analysis using a previously identified transcriptomic network (MMNet) found to be important in the context of macrophage nucleation. Specifically, the authors integrate the MMNet with human eBMD GWAS data and results from a large-scale mouse phenotyping effort to identify novel genes impacting osteoclast function and bone mass. All reviewers thought the manuscript was well-written and an outstanding example of the power of systems genetics. The phenotypic assessment of novel genes using large-scale mutant mouse phenotyping scans was also viewed as a significant strength. However, the reviewers questioned whether this work provided direct evidence that the MMnet genes are bona fide regulators of osteoclast fusion/multinucleation and that few mechanistic details were provided. To improve the manuscript we recommend providing a more detailed analysis of the bone phenotypes (i.e. better analysis of osteoclasts (e.g. simply counting nuclei) and, to the extent possible, osteoblast contributions to the phenotype). It was suggested that a more detailed and balanced assessment of at least one of the mutant mice be provided, e.g. Slc40a1 is a good candidate given it was deleted in a macrophage-specific background. Below are the detailed comments from the reviewers.

Essential revisions:

1) The term "osteoclast multinucleation" is used throughout the manuscript, but it is not very accurate. In each of the gene suppression or gene deletion studies, they observed reduced formation of osteoclasts, but it is not clear at which stage the genes function – they may be involved in osteoclast precursor maturation (TRAP+ mononuclear cells have not been counted), or may be involved in initiation of binucleation. At no point have number of nuclei been counted. All that can be said at this point is that they are involved in osteoclast formation.

2) The authors state multiple times (particularly in the Introduction and Discussion section) that multinucleation capability correlates with resorptive activity, but this is not true. There are many conditions (e.g. Src deletion, Pyk2 deletion, ClC-7 deficiency – the so-called "osteoclast-rich osteopetroses") where multinucleation occurs but osteoclasts are non-functional (or have impaired function).

3) In the phenotypic analysis of the mice, the authors explain all phenotypes by a change in osteoclast generation with no analysis presented on the other side of the equation – osteoblast numbers and bone formation rate. This oversight is disappointing. It is not clear that the high bone mass phenotypes are truly osteopetrotic – no data on cartilage remnants is shown.

4) It is very concerning that the number of mice analysed by histomorphometry is a very small subset of those analysed by micro-CT (e.g. for Bcat1 it is n=4 rather than n=8, and for Slc40a1 it is 4-5 instead of n=8). These are very low sample numbers indeed, and for the Slc40a1 data it is not clear that any affect on osteoclast formation is observed due to the biological variance. How were the subset of samples chosen for histomorphometry? Why were 50% of the samples excluded?

5) The description of osteoclastogenesis assays is very limited. Although osteoclast formation is suppressed by the siRNA experiments, there is no information presented about whether it is truly an effect on "multinucleation" or an effect on precursor differentiation. Was there any suppression in the number of TRAP+ mononuclear cells? Were binucleated cells observed? Did the osteoclasts formed contain less nuclei than the control osteoclasts.

6) siRNA is used to assess the affect of gene suppression on osteoclast formation, but no evidence is presented to show success of the knockdown.

7) Images shown in Figure 1E and Figure 3B for si-scrambled and si-DCSTAMP are identical. Were all these analyses performed together? If so, the data should not be presented twice.

8) Although deletion of Bcat1, Pik3cb, Atp8b2, Slc40a1, and Appl2 all lead to reduced osteoclast formation and increased BV/TV, the phenotypes are different in their effects on strength and cortical bone. This is not discussed.

9) The authors analyse knockout mice of 12 candidate co-regulatory genes with modest to mild bone phenotypes, some of which have been previously described in detail and implicated in osteoclast function and bone mass including Pik3cb, Deptor and Slc40a1. With the exception of Slc40a1, all of the bone parameters assessed arise from global or heterozygous (Atp8b2) knockout mice from a single age-point/sex (i.e. 12-wk old male Bcat1 and others 16-week old female) making it difficult to assess the penetrance and robustness of bone phenotype across both sexes and whether the phenotype resolves or increases with age as is frequently observed in the bones of mice harbouring dysfunctional osteoclasts. Moreover, changes in femur length observed (e.g. for Igsf8 and WKY rats) imply that bone phenotype(s) are not restricted to osteoclasts but likely extends to other bone/cartilage resident cell types such as chondrocytes or osteoblasts and/or osteocytes. However, the authors make no provisions to address cell-autonomous defects experimentally nor has this been acknowledged in the Discussion section, which misleads the reader. Can the authors lend any information as to whether the phenotypes are maintained across both sexes and time? Further, are the MMnet genes expressed and/or functionally implicated in other bone/cartilage resident cells? As a minimum this needs to be addressed in the Discussion section and statements implicating osteoclasts tempered accordingly.

10) Similarly, the notion that the low bone mass phenotype and shorter bones in WKY rats reflects disturbances in osteoclast multinucleation is unsubstantiated by the data presented in Figure 2—figure supplement 1. There has been little attempt to address this at either at the histological/histomorphometric level or ex vivo generation of osteoclasts which is well within their remit. Do the authors have any evidence that osteoclast numbers and/or fusion rates are increased in WKY? I can only identify data pertaining to multinucleated giant cells in the study of Kang et al., 2014. This data is important as it underscores the premise of the MMnet network from which ensuing target fusion genes and functional studies on osteoclasts are derived. As an aside, caution should be exercised when drawing direct comparisons between multinucleation regulatory networks of macrophage-derived giant cells and osteoclasts given that they are enzymatically and functionally distinct polykaryons.

11) In their present form, the in vitro studies remain too premature to draw meaningful correlations with in vivo data and thus fail to support conclusions that these genes regulate osteoclast fusion and/or resorption. Given the ready access to the IMPC mice it is surprising that the authors did not capitalize on this resource to validate cell-autonomous defects on osteoclast formation and multinucleation parameters ex vivo for at least a subset of the identified MMnet genes. Such data would go a long way towards substantiating claims and strengthening the resolve of the human osteoclast data, which are derived from siRNA-mediated transient knockdown experiments for which the efficiency and specificity of targeting remains incomplete for many of the described MMnet genes. This is of particular importance for the Slc40a1(Ferroportin) gene given the claims that " "Slc40a1 acts as an important determinant of adult bone mass and…. by directly regulating osteoclast multinucleation and function.", and in light of the discrepancies observed between the bone phenotypes in this study with those detailed by Wang et al., 2019, who reported reduced bone mass upon Slc40a1 deletion and accelerated osteoclast differentiation/fusion in the same LysoM-Cre conditional knockout model.

12) Although collectively the data presented in this study encompass a nice synergy between genetic and high through bone phenotyping, individually the data offers few mechanistic insights into the roles of the identified multinucealation factors in osteoclasts. Given the robust inhibition of osteoclast multinucleation following Slc40a1, any additional data here would be welcomed to strengthen is potential role a regulator of osteoclast fusion.

---

## [Author Response]

Summary:This manuscript describes a systems genetics analysis using a previously identified transcriptomic network (MMNet) found to be important in the context of macrophage nucleation. Specifically, the authors integrate the MMNet with human eBMD GWAS data and results from a large-scale mouse phenotyping effort to identify novel genes impacting osteoclast function and bone mass. All reviewers thought the manuscript was well-written and an outstanding example of the power of systems genetics. The phenotypic assessment of novel genes using large-scale mutant mouse phenotyping scans was also viewed as a significant strength. However, the reviewers questioned whether this work provided direct evidence that the MMnet genes are bona fide regulators of osteoclast fusion/multinucleation and that few mechanistic details were provided. To improve the manuscript we recommend providing a more detailed analysis of the bone phenotypes (i.e. better analysis of osteoclasts (e.g. simply counting nuclei) and, to the extent possible, osteoblast contributions to the phenotype). It was suggested that a more detailed and balanced assessment of at least one of the mutant mice be provided, e.g. Slc40a1 is a good candidate given it was deleted in a macrophage-specific background. Below are the detailed comments from the reviewers.Essential revisions:1) The term "osteoclast multinucleation" is used throughout the manuscript, but it is not very accurate. In each of the gene suppression or gene deletion studies, they observed reduced formation of osteoclasts, but it is not clear at which stage the genes function – they may be involved in osteoclast precursor maturation (TRAP+ mononuclear cells have not been counted), or may be involved in initiation of binucleation. At no point have number of nuclei been counted. All that can be said at this point is that they are involved in osteoclast formation.

We thank the reviewer for highlighting this important point and agree it is not clear in the text that the number of nuclei in each differentiated mature osteoclast was determined (see also our response to comment 5). We also agree with the reviewer that osteoclast differentiation and fusion/multinucleation are two distinct cellular stages and any gene silencing experiment should take this into account. We now clarify that all RNAi experiments were performed at Day 3, in RANKL-differentiated and TRAP positive mononucleated osteoclasts, to assess the effect of gene silencing on cell multinucleation/fusion *per se* at day 6. This experimental design allows working on fully differentiated TRAP+ osteoclasts and focus on cell fusion/multinucleation during a 48-hour period.

We have updated the Materials and methods section, text and figure legends to clarify the experimental design. Furthermore, in accordance with comment 5, we now include the percentage of TRAP+ mononuclear, binuclear and trinuclear osteoclasts following si-RNA for each gene (Figure 4—figure supplement 2 Panel E and response to comment 5).

2) The authors state multiple times (particularly in the Introduction and Discussion section) that multinucleation capability correlates with resorptive activity, but this is not true. There are many conditions (e.g. Src deletion, Pyk2 deletion, ClC-7 deficiency – the so-called "osteoclast-rich osteopetroses") where multinucleation occurs but osteoclasts are non-functional (or have impaired function).

We thank the reviewer for highlighting this omission and we have amended the introduction accordingly and added an additional reference to ensure accuracy and clarity throughout. (Introduction).

3) In the phenotypic analysis of the mice, the authors explain all phenotypes by a change in osteoclast generation with no analysis presented on the other side of the equation – osteoblast numbers and bone formation rate. This oversight is disappointing. It is not clear that the high bone mass phenotypes are truly osteopetrotic – no data on cartilage remnants is shown.

We are very grateful to the reviewer for raising this important point. As suggested by the reviewer we quantified the fraction of retained cartilage in both trabecular and cortical bone in distal femur sections from (i) *Bcat1^-/-^* and WT control mice and (ii) *Slc40a1*^∆lysMCre^ and *Slc40a1*^flox/flox^ controls by staining with Alcian Blue Van Geison. Consistent with impaired osteoclast function in these mutant mice we observed increased retained cartilage in both trabecular and cortical bone in *Bcat1^-/-^* mice and increased retained cartilage in trabecular bone in *Slc40a1^-/-^* mice. We have added these data to Figure 2—figure supplement 1 and Figure 5—figure supplement 2 and have updated the Materials and methods section, Results section and figure legends accordingly.

4) It is very concerning that the number of mice analysed by histomorphometry is a very small subset of those analysed by micro-CT (e.g. for Bcat1 it is n=4 rather than n=8, and for Slc40a1 it is 4-5 instead of n=8). These are very low sample numbers indeed, and for the Slc40a1 data it is not clear that any affect on osteoclast formation is observed due to the biological variance. How were the subset of samples chosen for histomorphometry? Why were 50% of the samples excluded?

As requested by the reviewers, we have now included histomorphometry data from the full set of n=8 samples for *Bcat1^-/-^* and *Slc40a1^-/-^* mice (see revised Figure 2 and Figure 5).

5) The description of osteoclastogenesis assays is very limited. Although osteoclast formation is suppressed by the siRNA experiments, there is no information presented about whether it is truly an effect on "multinucleation" or an effect on precursor differentiation. Was there any suppression in the number of TRAP+ mononuclear cells? Were binucleated cells observed? Did the osteoclasts formed contain less nuclei than the control osteoclasts.

We thank the reviews for highlighting this important issue. As discussed in point 1 we now clarify that the number of nuclei in each differentiated mature osteoclast were determined following RNAi. This comprehensive analysis enables determination of the consequences of si-RNA knockdown on osteoclast multinucleation/fusion in addition to cell differentiation and maturation. Accordingly, we have revised Figure 4—figure supplement 1 to include the percentage of TRAP+ mono, bi and trinucleated osteoclasts following si-RNA. We have updated the methods, the text and the figure legends to clarify the experimental design.

6) siRNA is used to assess the affect of gene suppression on osteoclast formation, but no evidence is presented to show success of the knockdown.

Quantification of gene knockdown following si-RNA was previously included in Figure 4—figure supplement 1 and referred to in subsection “Human MMnet orthologues regulate osteoclast multinucleation and resorption”. We have now emphasised these data clearly in the text.

7) Images shown in Figure 1E and Figure 3B for si-scrambled and si-DCSTAMP are identical. Were all these analyses performed together? If so, the data should not be presented twice.

We thank the reviews for highlighting this error we have now amended Figure 2 accordingly.

8) Although deletion of Bcat1, Pik3cb, Atp8b2, Slc40a1, and Appl2 all lead to reduced osteoclast formation and increased BV/TV, the phenotypes are different in their effects on strength and cortical bone. This is not discussed.

We thank the reviewers for highlighting this important point. We now consider this point in detail and have amended the Discussion section.

9) The authors analyse knockout mice of 12 candidate co-regulatory genes with modest to mild bone phenotypes, some of which have been previously described in detail and implicated in osteoclast function and bone mass including Pik3cb, Deptor and Slc40a1. With the exception of Slc40a1, all of the bone parameters assessed arise from global or heterozygous (Atp8b2) knockout mice from a single age-point/sex (i.e. 12-wk old male Bcat1 and others 16-week old female) making it difficult to assess the penetrance and robustness of bone phenotype across both sexes and whether the phenotype resolves or increases with age as is frequently observed in the bones of mice harbouring dysfunctional osteoclasts. Moreover, changes in femur length observed (e.g. for Igsf8 and WKY rats) imply that bone phenotype(s) are not restricted to osteoclasts but likely extends to other bone/cartilage resident cell types such as chondrocytes or osteoblasts and/or osteocytes. However, the authors make no provisions to address cell-autonomous defects experimentally nor has this been acknowledged in the Discussion section, which misleads the reader. Can the authors lend any information as to whether the phenotypes are maintained across both sexes and time? Further, are the MMnet genes expressed and/or functionally implicated in other bone/cartilage resident cells? As a minimum this needs to be addressed in the Discussion section and statements implicating osteoclasts tempered accordingly.

Skeletal samples from mice, harbouring global deletion of the 12 candidate co-regulatory genes, were examined as part of The Origins of Bone and Cartilage Disease (OBCD) Programme and International Mouse Phenotyping Consortium (IMPC). Details of these programmes are referenced in subsection “Animals”. In the IMPC programme all mice are sacrificed at 16 weeks of age and skeletal phenotypes determined in female samples (n=2-6 per line) and joint phenotypes determined in male samples (n=2-6 per line) by the OBCD programme. Thus, detailed skeletal phenotype data is only available from 16 week old female mice and not from males or older animals. Nevertheless, the broad IMPC screen also identified increased bone mineral content by DEXA in male Atp8b2^+/-^ mice at 16 weeks of age. We agree with the referees that since these mice are global knockouts the observed skeletal phenotypes may also reflect the consequences of gene deletion in non-monocyte/macrophage lineages. The Discussion section has been amended accordingly.

10) Similarly, the notion that the low bone mass phenotype and shorter bones in WKY rats reflects disturbances in osteoclast multinucleation is unsubstantiated by the data presented in Figure 2—figure supplement 1. There has been little attempt to address this at either at the histological/histomorphometric level or ex vivo generation of osteoclasts which is well within their remit. Do the authors have any evidence that osteoclast numbers and/or fusion rates are increased in WKY? I can only identify data pertaining to multinucleated giant cells in the study of Kang et al., 2014. This data is important as it underscores the premise of the MMnet network from which ensuing target fusion genes and functional studies on osteoclasts are derived. As an aside, caution should be exercised when drawing direct comparisons between multinucleation regulatory networks of macrophage-derived giant cells and osteoclasts given that they are enzymatically and functionally distinct polykaryons.

The reviewers have raised an important point; to address it we have performed osteoclast cultures from WKY and LEW rats (n=3/group). We cultured rat osteoclasts for 6 days in the presence of M-CSF and RANKL and found relatively larger osteoclasts with a greater number of nuclei in WKY rats. This increased multinucleation/fusion was also associated with increased expression levels of two key genes implicated in osteoclast fusion: Trem2, the master-regulator of MMnet and Nfatc1, the master-regulator of transcriptional reprogramming of osteoclast fusion. These data are now included in the revised version of Figure 1 and the text/figure legends changed accordingly.

11) In their present form, the in vitro studies remain too premature to draw meaningful correlations with in vivo data and thus fail to support conclusions that these genes regulate osteoclast fusion and/or resorption. Given the ready access to the IMPC mice it is surprising that the authors did not capitalize on this resource to validate cell-autonomous defects on osteoclast formation and multinucleation parameters ex vivo for at least a subset of the identified MMnet genes. Such data would go a long way towards substantiating claims and strengthening the resolve of the human osteoclast data, which are derived from siRNA-mediated transient knockdown experiments for which the efficiency and specificity of targeting remains incomplete for many of the described MMnet genes. This is of particular importance for the Slc40a1(Ferroportin) gene given the claims that " "Slc40a1 acts as an important determinant of adult bone mass and…. by directly regulating osteoclast multinucleation and function.", and in light of the discrepancies observed between the bone phenotypes in this study with those detailed by Wang et al., 2019, who reported reduced bone mass upon Slc40a1 deletion and accelerated osteoclast differentiation/fusion in the same LysoM-Cre conditional knockout model.

The reviewers raise an important question. Rapid-throughput skeletal phenotyping of IMPC mutant mice was performed as part of the Wellcome Trust funded Origins of Bone and Cartilage Disease Programme (Bassett et al., 2012; Kemp et al., 2017; Medina-Gomez et al., 2017; Morris et al., 2018; and reviewed in Freudenthal et al., 2018). In this programme, mice were generated at the Sanger Institute and only skeletal samples provided for phenotype analysis. Importantly, once samples are collected at 16 weeks of age colonies of IMPC mice are not maintained and are only archived as embryo’s or sperm. Thus, we are unable to perform primary osteoclast cultures from the 12 IMPC knockout lines.

Furthermore, in Figure 4—figure supplement 1 (Panel B) we now show that the efficiency of si-RNA knockdown was greater than 80% for all genes studied in human osteoclasts.

Importantly, the RNAi studies in human osteoclasts demonstrated that changes in multinucleation correlated with resorption in vitro and trabecular bone volume in vivo (Figure 4E and F).

With regard to studies in *Slc40a1* mutant mice, *Slc40a1* encodes the mammalian iron transporter (ferroportin) and its deficiency in the myeloid lineage is predicted to have two main consequences; (i) intracellular ferrous (Fe^2+^) iron levels will increase because of a defect in export. This is likely to impair osteoclast activity, fusion and multinucleation as these are energy-demanding processes that require appropriate levels of intracellular iron, and (ii) defective iron export in myeloid cells is predicted to reduce iron concentrations in the extracellular bone microenvironment and may result in defective osteoclast-osteoblast coupling.

We are grateful that the editor agreed that additional studies of these cell-intrinsic and osteoclast-osteoblast coupling pathways were beyond the scope of the current manuscript, which is focused on the validation of a *trans*-eQTL network regulating osteoclast multinucleation and bone mass.

12) Although collectively the data presented in this study encompass a nice synergy between genetic and high through bone phenotyping, individually the data offers few mechanistic insights into the roles of the identified multinucealation factors in osteoclasts. Given the robust inhibition of osteoclast multinucleation following Slc40a1, any additional data here would be welcomed to strengthen is potential role a regulator of osteoclast fusion.

We agree with the reviewers that additional mechanistic data relating iron metabolism to the regulation of osteoclast multinucleation would enhance these studies but we are grateful to the editor for agreeing that such additional experiments are beyond the scope of the current manuscript.